# The psychosocial determinants of adherence to home-based rehabilitation strategies in parents of children with cerebral palsy: A systematic review

Japhet Niyonsenga[1,2]*, Liliane Uwingeneye[3], Inès Musabyemariya[4], Jean Baptiste Sagahutu[5], Francesca Cavallini[2], Luca Caricati[2], Rutembesa Eugene[1], Jean Mutabaruka[1], Stefan Jansen[1], Nadia Monacelli[6]

1 Department of Clinical Psychology, College of Medicine and Health Sciences, University of Rwanda, Kigali, Rwanda, 2 Department of Social Sciences and Humanities, University of Parma, Parma, Italy, 3 Department of Business Administration, College of Business and Economics, University of Rwanda, Kigali, Rwanda, 4 Department of Physiotherapy, International Committee of the Red Cross–ICRC, Tinduf, Algeria, 5 Department of Physiotherapy, College of Medicine and Health Sciences, University of Rwanda, Kigali, Rwanda, 6 Department of Economy, University of Parma, Parma, Italy

* niyonsengajaphet74@gmail.com

**Data Availability Statement:** All relevant data are within the manuscript and its Supporting information files.

## Abstract

### Introduction

Involving parents of children with cerebral palsy (C-CP) in home exercise programmes (HEP) is globally practiced strategy closely linked to improved physical performance and functional outcomes for the child. Nevertheless, non-adherence to HEP is increasing at an alarming rate, and little is known about the factors influencing adherence to HEP (AHEP) especially in parents of C-CP. This systematic review aimed to identify the factors enhancing AHEP among parents of C-CP to reinforce the efficacy of rehabilitation practices proposed by health professionals, researchers, and educators.

### Materials and methods

We conducted searches in PubMed, Scopus, CINHAL, PsycINFO, and Embase for articles published up to March 2023, that investigated the factors influencing AHEP among parents of C-CP. A narrative synthesis was conducted using the search results and pertinent material from other sources.

### Results

Overall, non-adherence rates to HEP were moderate to high, ranging from 34% to 79.2%. Strong evidence suggests that factors enhancing AHEP fall into three categories: child-related (such as younger age and better gross motor function [GMF]), the caregiver-related (including high self-efficacy and knowledge, strong social support, low levels of depression, anxiety and stress symptoms, and a low perception of barriers), and the physiotherapist-

**Funding:** The author(s) received no specific funding for this work.

**Competing interests:** The authors have declared that no competing interests exist.

related. For the latter category, the parent's perception of a supportive and collaborative relationship with the therapist is one of the conditions most favourably influences AHEP.

## Conclusion

Our findings highlight that factors influencing AHEP are multifactorial. Some, such as GMF or the economic and social conditions of the family, are challenging to change. However, the relationship between therapist and parent is an aspect that can be strengthened. These results underscore the importance of substantial training and psychosocial support for therapists to enhance their awareness and competence in building supportive relationship with parents.

## Introduction

Cerebral palsy (CP) represents the largest diagnostic group treated in paediatric rehabilitation [1, 2]. It is characterized by a spectrum of motor development issues stemming from persistent damage to the immature brain [3]. The most common manifestations of CP include deficits in sensory, communicative, cognitive, and musculoskeletal functions, significantly limiting children's autonomy in daily activities. [1, 2, 4] Approximately 60% of children with CP face challenges with the effective arm and hand use for reaching, grasping, releasing, and manipulating objects which restricts their ability to perform daily tasks [1]. Additionally, CP is associated with several comorbidities, including epilepsy, presents in 35%-66.36% cases [5, 6], speech impairment (43.5%), visual impairment (25%-58%) [6, 7], and cognitive impairment (24%) [6].

Data from national cerebral palsy registries and population-based studies in Europe, Australia, and the United States indicate a CP prevalence of approximately 1.8–2.3 per 1000 children [8–10]. However, in Africa, the prevalence may be as high as 2–10 per 1000 children [11, 12]. with the main contributing factors in high-income countries being premature or low birth weight [13], and in Africa, factors include birth asphyxia, kernicterus, central nervous system infections and congenital and neonatal infections which are more readily preventable [5, 12].

Although the consequences of CP are permanent, early intervention programs can significantly improve children's independence and quality of life. These programs are most effective when implemented between birth and five years of age, a period marked by heightened neuroplasticity [14–19]. Early intervention enables children with cerebral palsy (C-CP) to enhance their potential abilities by addressing initial movement limitations. The success of rehabilitation also heavily relies on consistent and proper home exercise programs (HEP) which serve as an extension of treatment. In the paediatric context, HEP involves training parents or caregivers by medical professional on executing the prescribed exercises at home and between physical therapy sessions [17, 20].

HEP can constitutes 50–80% of total therapy received by C-CP [17] and has been shown to not only benefit the children but also increase parental satisfaction, according to Novak [21]. Furthermore, HEP is crucial for families as it saves time and money, involves parents in treatment decisions and allows them to choose the best options for their child [1, 22].

Nevertheless, Adherence to HEP (AHEP) poses a significant challenge in rehabilitation, with non-adherence rates estimated between 50% and 66% [23, 24], expectedly higher in resource-poor settings [23]. Parental non-adherence adherence has been identified primary cause for treatment failure in long-term paediatric conditions [23] and is linked to unnecessary

and detrimental alterations to treatments prescribed by healthcare professionals. Given these issues, a systematic review to investigate the factors determining parental adherence to HEP is warranted. Recent systematic review focusing on predictors of adult patients' adherence to home-based physical therapies have identified intention to engage in home-based physical therapy, self-motivation, self-efficacy, previous adherence to exercise-related behaviours, and social support as strong predictors of adherence [25]. Similarly, another review examining barriers to treatment adherence in physiotherapy outpatient clinics found that poor treatment adherence was generally associated factors such as low levels of physical activity in previous weeks, low self-efficacy, depression, anxiety, feelings of helplessness, inadequate social support, a greater perceived number of barriers to exercise, and increased pain levels during exercise among patients undergoing muscle-skeleton physiotherapy [26].

Despite these insights, there has been minimal focus on understanding of the factors influencing the adherence to HEP among parents of C-CP, and to date, no systematic review has specifically addressed this issue. Recognizing this research gap, the current study aims to provide a comprehensive overview of the factors that influence parents' compliance with HEP for their C-CP. This area of study is crucial as understanding the barriers to adherence can enable clinicians to identify parents who may be at risk of non-adherence and develop strategies to mitigate these barriers, thereby maximising adherence. By addressing these challenges, the study seeks to highlight effective strategies for enhancing the implementation of home-based rehabilitation, which is a vital component of achieving optimal clinical outcomes for C-CP.

## Materials and methods

### Study design

This systematic review follows the Preferred Reporting Items for Systematic Review and Meta-Analysis Protocols (PRISMA). The PRISMA guidelines provide items informed by empirical research for systematic reviews and meta-analyses [27].

### Search strategy and selection criteria

The sources of information that were used are Medline through PubMed, Scopus, Cumulative Index to Nursing and Allied Health Literature (CINHAL), PsycINFO, and EMBASE, covering the period from database inception to March 15, 2023. The search strategies for this review were established by applying Boolean operators 'AND' and 'OR' to effectively connect the descriptors, facilitating a focused yet exhaustive search. These descriptors were derived from keywords and MeSH terms search (in PubMed) in each database's metadata system. Therefore, the strategy was carefully developed for each database, considering its unique indexing system and search functionalities to guarantee the identification of the most pertinent and extensive collection of literature.

The search terms covered a spectrum of terms relating to home exercise programs, adherence, diverse influencing factors and cerebral palsy. To ensure the inclusion of a wide range of studies, the search incorporated terms with prefixes. The sample query for PubMed was constructed as follows: ("exercise" OR "therap*" OR "physiotherap*" OR "home physiotherap*" OR "home exercise*" OR "rehabilitation exercise*" OR "physical therapy") AND ("adheren*" OR "complian*" OR "involvement") AND ("psychosocial factor*" OR "psychological factors" OR "motivat*" OR "barrier*" OR "facilitat*") AND ("cerebral pals*" OR "neurodevelopmental disorder*" OR "pediatric neurology").

Two separate researchers independently conducted the search and evaluated studies for eligibility based on the inclusion criteria (JN, LU) using the Covidence database (www.covidence.org). During instances of disagreement, a dedicated reviewer panel (a team of MN,

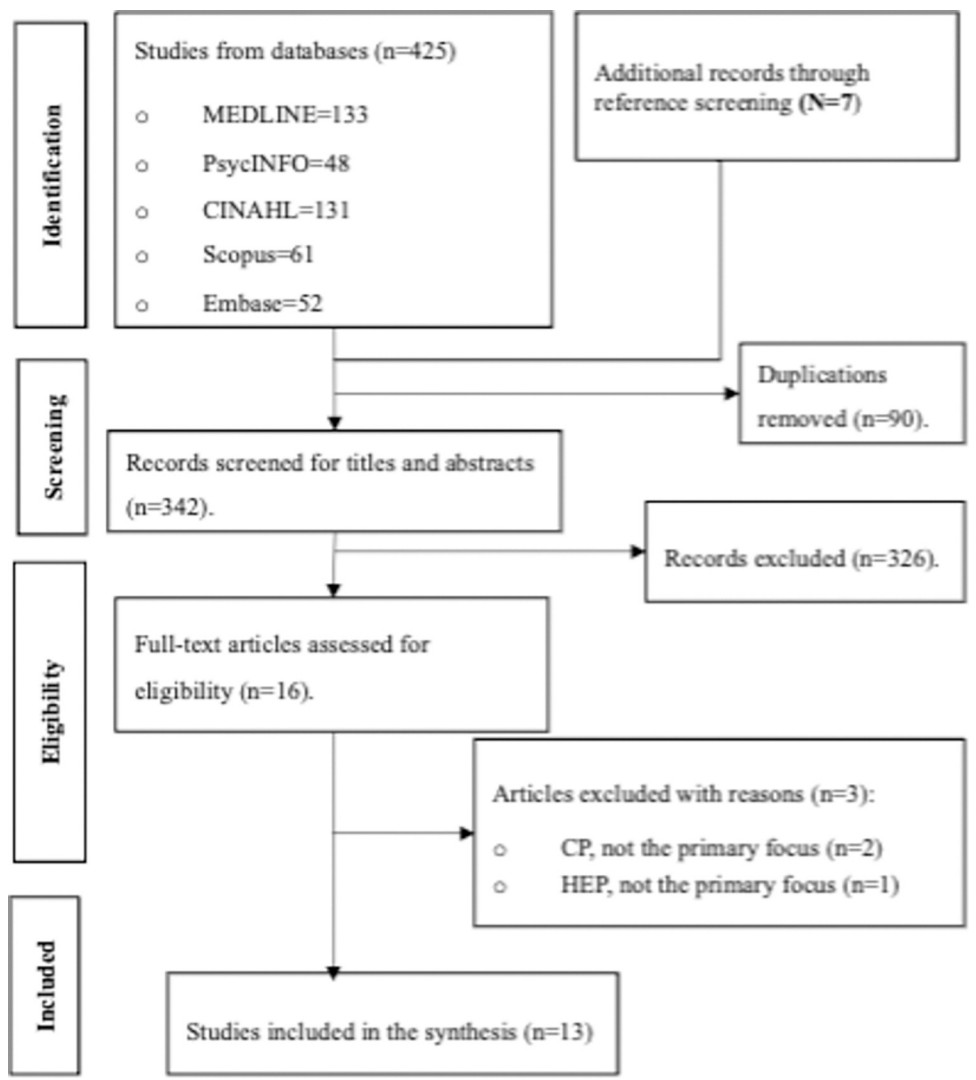

**Fig 1. "Flow chart of study selection" should be placed here.**

LC, and FC) was engaged to achieve consensus. Studies were considered eligible for inclusion if they (i) involved parents of children with CP who were younger than 18 years; (ii) described exercises that were assisted by a caregiver rather than performed independently by the patient; and (iii) were published in English. Notably, studies that presented both clinic and home-based exercise adherence data were included, provided the home-based data were reported separately. Exclusions were made for (i) review articles; (ii) studies involving children without a confirmed CP diagnosis; (iii) studies not available in full text; and (iv) computer-based rehabilitation studies due to a lack of reported data on psychosocial determinants of adherence from parents' perspectives, which was a critical focus of our review.

The selection process began with the Boolean combination of descriptors, followed by a review of titles, abstracts, and finally, full texts. All references were imported into Covidence for deduplication and screened for eligibility based on the established criteria. The number of articles identified through electronic databases and the reasons for exclusion were rigorously documented in Fig 1. Following identification of relevant papers, their reference lists were

manually searched for additional pertinent literature. At the conclusion of the selection phase, chosen papers were organized for data extraction, tabulation, and analysis, and subsequent interpretation and reporting.

## Data extraction and management

The study characteristics and outcome data were extracted by one reviewer using a pre-designed data extraction spreadsheet. The information extracted for analysis was divided into three categories: study data, data about children with CP, and data about the parents. The following study details were extracted: study title, author(s), publication year, country of study, study design, participant characteristics (**e.g.**, child's age range, parents' mean age) and the measure of adherence to the home exercise program. If some of the articles did not contain all the data required for the review, we included the available data (Table 1).

## Quality assessment

For quantitative studies, we utilized the Newcastle-Ottawa Scale (NOS) adapted for cross-sectional studies. This decision was informed by the nature of the review studies, which were explicitly of this type [28]. The NOS evaluates the methodological quality of studies based on three critical domains: Selection, comparability, and outcome. Stars are awarded in these categories according to the guidelines provided in the NOS coding handbook, with up to five stars might be given for selection, two for comparability, and three for outcome. The total number of stars a study receives serves as an indicator of its quality.

In assessing the qualitative studies, the Critical Appraisal Skills Programme Tool (CASP) for Qualitative Research. The tool encompasses 10 criteria aimed at appraising a study's methodological rigour and validity [29]. We adopted a rating approach to gauge the relative quality of these studies [30]. Those meeting at least eight criteria were deemed 'high quality', those fulfilling 5–7 criteria were considered 'moderate quality' if they met 5–7 of the criteria, and studies meeting four or fewer criteria were categorized as 'poor quality'.

## Data synthesis

The first plan was to do a meta-analysis. Unfortunately, there was little data provided in the studies found by this review, and the data that was gathered was inconsistent in terms of measures employed (7 out of 9 were self-designed) and analysis undertaken. As such, it was decided to conduct a narrative synthesis rather than a meta-analysis to synthesize the data in the most meaningful way possible, guided by the concepts described by Popay and colleagues [31]. A narrative synthesis approach was used to describe, compare, and combine findings from multiple studies.

This method concentrates on textual summaries and results descriptions with the primary goal of "telling the tale "of the findings from the included research [31]. The elements of this process in this review included: developing a preliminary synthesis to organize the findings from the included studies; exploring relationships within this data to understand and explain patterns or differences; and assessing the robustness of the synthesis by considering the strength, quality, and generalizability of the included data.

## Results

### Characteristics of included studies

The search yielded 432 records. After removing duplicates, we screened a total of 342 titles and abstracts, leading to the exclusion of 326 records due to irrelevance. We then assessed the full

**Table 1. Summary of the main study characteristics and the key findings.**

| Authors | Year | Sample details | Non-adherence Rate | child's age | Concept (and measure) of Adherence to HEP | Key findings | Study quality (NOS/ CASP) |
|---|---|---|---|---|---|---|---|
| BAŞARAN et al. [34] | **2014** | 147 parents (43.5% females, mean age [MA] = 34.3),Turkey | 52.9% | 2–18 years | self-designed questionnaire with three close-ended questions. | Gross motor function classification system (OR = 1.50, p = 0.01); Emotional exhaustion (OR = 0.92, p<0.001); Depression and anxiety were not associated with adherence. | NOS-7 stars |
| Alwhaibi et al. [22] | 2022 | 109 parents (99 females, MA = 33.7, Saudi Arabia | 66.10% | Below 12 years | Adapted questionnaire composed of 17 items scored on a 5-point Likert scale. | Social support (β = 0.156, p = 0.041); Health professionals' behaviour (β = 0.268, p = 0.026); Mother's sense of unsureness (β = -0.306, p = 0.003) | NOS-8 stars |
| Lillo-Navarro et al. [37] | 2019 | 219 parents (85% female; MA = 34.3), Spain | 54.2% | Not specified | Adherence assessed on a 5-point Likert scale. | Education level (OR = 0.40, p< 0.05); Child sitting abilities (OR = 2.12, p<0.05); low barriers perception (OR = 2.7, p<0.001); knowledge and ability (OR = 5.68, p<0.001); Giving information about evolution (OR = 6.27, p<0.05); Self-efficacy = 3.28, p<0.001); Justifying usefulness of exercises (OR = 9.49, p<0.001); Using the child as a model (OR = 2.72, p<0.05); Asking about adherence at home (OR = 2.98, p<0.001); Parents' high satisfaction with the care received (OR = 1.71, p<0.05) | NOS-9 stars |
| Medina-Mirapeix et al. [35] | 2017 | 219 parents (8 males, 99 females; MA = 34.3), Spain | 42.8% | 0.5–6 years | Adherence assessed on a five-point (1 = never, to 5 = always). | Barriers perception (OR = 5.68, p<0.001); High knowledge and ability (OR = 2.18, p<0.001); High self-efficacy (6.82, p<0.001); Social support (OR = 2.04, p<0.001); Age <2 years (OR = 1.99, P<0.05); Child inability to sit (OR = 2.23, p<0.05); Being in treatment <2 years (OR = 2.3, p<0.05); Justifying usefulness of exercises (OR = 2.17, p<0.05); Advising to insert HEP into daily (OR = 2.54, p<0.001); Checking skills (OR = 2.21, p<0.05). Asking about adherence at home (OR = 2.20, p<0.001); Parents' high satisfaction with the care received (OR = 1.71, p<0.05) | NOS-9 stars |
| Rone-Adams et al. [24] | 2004 | 66 parents, South Florida | 66% | Below 18 years | Self-designed questionnaire of six close-ended questions. | Stress level (F = 4.417, p<0.039, R$^2$ = 0.065); Family problems (r = 0.345, p = 0.005). | NOS-6 stars |
| Sel et al. [38] | 2020 | 155 parents, Turkey | 47.1% | 2–18 years | 28-items Validated Questionnaire measuring adherence. | Depression and anxiety (r = -0.405, p<0.001); Medication use (r = -0.233, p = 0.003); Device use (r = 0,741, p<0.001); Health professional attitudes (r = 0.221, p<0.001); Gross motor function (p<0.01). | NOS-7 stars |
| Tahayneh et al. [39] | 2020 | 48 mothers, **Palestine** | 35.4% | 1–3 years | Self-reported questionnaire was used to assess adherence on a 5-point Likert scale | Socio-economic, work, age, education and marital status of mothers were not associated with HEP. | NOS-6 stars |
| Gmmash et al. [32] | 2021 | 446 participants (96% females), **USA** | Not available | Below 5 years | A self-reported questionnaire was used to assess adherence | Therapists' modelling (r = 0.5, p<0.001); parent's self-efficacy (r = 0.6, p < .000); Child enjoyment (r = 0.5 p < .0001); Daily routines appropriateness (r = 0.7, p < .000); HA appropriate for home environment (r = 0.6, p < .000); Sibling and extended family inclusion (r = 0.3, p < .000). | NOS-7 stars |

(*Continued*)

**Table 1.** (Continued)

| Authors | Year | Sample details | Non-adherence Rate | child's age | Concept (and measure) of Adherence to HEP | Key findings | Study quality (NOS/CASP) |
|---|---|---|---|---|---|---|---|
| Galil et al. [40] | 2001 | 193 participants, **Israel** | 79.2% of Jews | 0.6–6 years | Compliance was measured by two questions from the validated questionnaire. | Positive adherence factors: Being Bedouin, parents' education, hopefulness, belief in exercise benefits, service satisfaction, socioeconomic status; Negative adherence factor: Questioning intensity destiny | NOS-7 stars |
| Tétreault et al. [41] | 2003 | 45 participants (90.3% females) **Canada** | 31.1% | 6–8 years | Self-reported questionnaire. | Positive adherence factors: Social support, motivation, hope, confidence, being more realistic, satisfaction with HEP. Negative adherence factors: Difficulty integrating exercise into daily life, feelings of burden, guilt, and discouragement. | NOS-6 stars |
| Louka-Lazouri et al. [42] | 2020 | 63 parents (51% female) **Greece** | 60% | 1–6 years | Self-reported questionnaire | Positive adherence factors: simple and fun exercises, therapeutic support, communication and understanding, parent training, and physiotherapy involvement. Negative adherence factors: fear of causing pain and negative views of physiotherapy. | NOS-6 stars |
| Demeke et al. [43] | 2023 | 17 participants (27–39 years), Ethiopia | Not available | 1–4 years | Individual, face-to-face, and semi-structured interviews | Negative adherence factors: Low family support, limited recourse, lack of knowledge and poor attitude towards home-based therapy. | CASP-10 stars |
| Lillo-Navarro et al. [44] | 2019 | 28 participants, Spain | Not specified | 0.5–6 years | Focus group discussion | Key Adherence themes: HEP characteristics (perceived effects, complexity, and number of exercises); and physiotherapist's teaching style (building parents' confidence, daily routine integration, incentives and motivation). | CASP-10 stars |

**Notes**: Nos = The Newcastle-Ottawa Scale; CASP = the Critical Appraisal Skills Programme for Qualitative Research

texts of the remaining 16 records for eligibility, excluding three that did not meet the criteria. This process resulted in 13 relevant records. The flow chart of the literature search process is illustrated in **Fig 1**. We contacted the corresponding and last authors of these 13 records to inquire about any related studies, receiving responses from 8 (59%) authors who responded with either a suggestion or no additions at all, resulting in two additional records already included in the initial 13 records.

Our final review included eleven quantitative and two qualitative studies that explored factors influencing parents' AHEP (**Table 1**). These studies were conducted across various countries: three studies were conducted in Spain, two in Turkey, two in the USA, and one each in Saudi Arabia, Ethiopia, Israel and Palestine. Publication dates ranged from 2000 to 2023, with three studies published between 2000 and 2005, and the remaining 10 studies between 2013 and 2023. Although the samples in all studies were mixed, participants were predominantly female, making up between 51% to 99% of the total sample. Sample sizes varied significantly, from 17 to 446 participants. The average age of parents across was approximately 34 years, while the age range for children spanned from 0.5–18 years. Adherence measures relied on participant self-report, with results often dichotomized into 'adherent' and 'non-adherent' individuals for analysis or expressed as a percent of the recommended exercise done.

## Quality assessment of the included articles

In appraising the quantitative studies, two studies were awarded nine stars, one study was received eight stars, four were awarded seven stars, and the remaining four studies also

awarded seven stars (Table 1). The primary concern identified during the quality assessment was the universal use of convenience sampling across all studies, suggesting that participants might not accurately represent the target research population. Except for one study [32], none conducted power analyses or provided sample size estimates to justify their sample sizes. While there are no universal guidelines for determining the optimal sample size for multivariate analyses, a small sample size can undermine confidence in the findings [33]. However, given the nature of the target population, the use convenient sample is understandable, yet it necessitates the authors adapting appropriate their analyses to the quality of their sample. Consequently, variations in study sample sizes might account for the inconsistencies in significant findings regarding specific factor. Moreover, response rates were generally low across the studies, with no insights into the characteristics of non-responders. This omission makes assessing the representative of the participants challenging [25]. It is plausible that parents who were most adherent to HEP may have been more willing to participate in the research because they saw it as relevant and valuable. In contrast, non-adherent parents might have chosen not to discuss their experiences and abstain from the research [25].

A notably issue was the frequent use of non-validated measures for various concepts, and the reliability evaluations of the scales for the specific samples were either not consistently conducted [24, 34, 35] or did not consistently demonstrate adequate reliability. Furthermore, all studies relied on self-report methods to assess AHEP and its associated factors, which depends on individuals accurately recalling and reporting their exercises routines. This approach exposes the data to potential social desirability and memory biases, possibly leading to an overestimation of adherence. This observation aligns with a previous systematic review that found that all 61 identified self-report measures lacked psychometric validity [36]. The current review also noted that factors such as social support and self-efficacy were sometimes assessed using a single item derived from related validated scales, which may not effectively capture the target concepts [32, 35, 37]. Other research efforts have tried to mitigate these issues by developing new measures based on existing scales or theories in the scientific literature [22, 35, 37] or by pre-testing and piloting the scales.

Conversely, the two qualitative studies included in the review demonstrated high quality in their approach to participant recruitment, data collection, and analysis (**Table 1**). These studies adequately considered the role of the researcher and how they might influence the study results.

## Factors influencing the adherence to home exercise programs among parents of children with cerebral palsy

Our analysis identified several psychosocial and situational factors that have been considered in the literature to date, and were grouped into four conceptually related categories: Child related factors, caregiver-related factors, and therapist guidance and advice (**Fig 2**). We relied on the descriptive information supplied in the original articles to address our secondary research aim of identifying the specific factors that are perceived to be beneficial or detrimental in adhering to HEP among parents of C-CP.

### Child-related factors of HEPA

*Gross motor function.* Four quantitative studies showed an inverse relationship between gross motor function and HEPA [34, 35, 37, 38]. These studies evidenced that parents' adherence appeared to be higher when the gross motor function was low.

*Child age, gender, and weight.* Four quantitative studies conducted showed mixed findings on the association between child age and HEPA [34, 35, 37]. While two studies revealed no

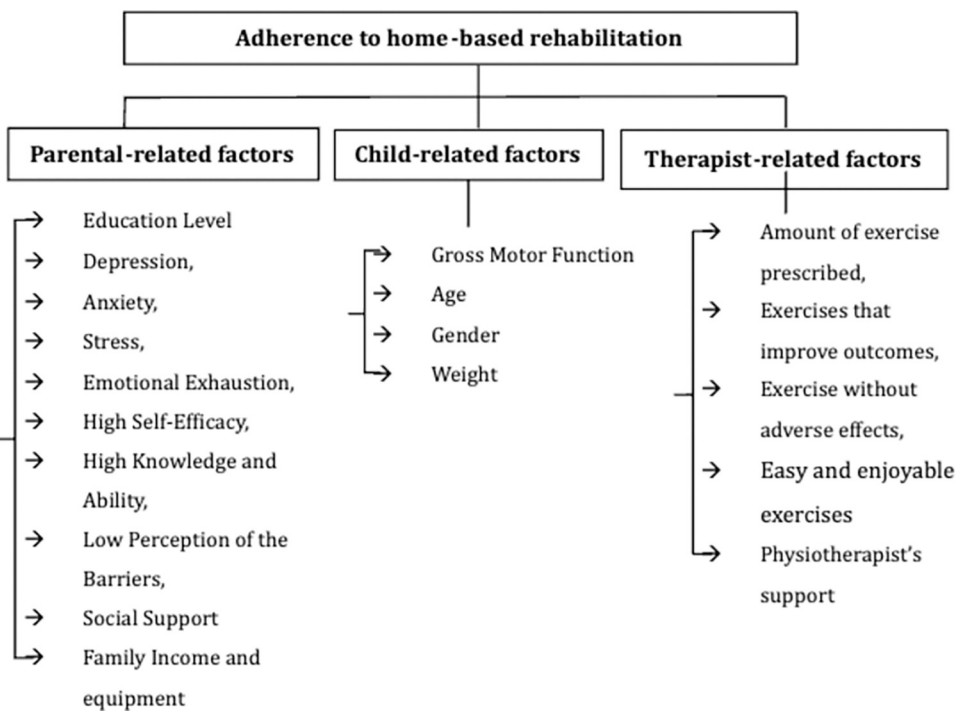

**Fig 2. "Factors influencing the adherence to home exercise programs among parents of children with cerebral palsy" can be placed here.**

associations between child age and HEPA [34, 37], one study showed a positive association, indicating that parents with younger children were more likely to adhere [35]. Two separate quantitative studies found no associations between the child's gender [37], the child's weight [34] and HEPA.

## Parental-related factors of HEPA

*Education level, age, and marital status.* Four quantitative studies have consistently revealed that mothers' educational status was positively related to adherence [24, 37, 40, 42]. These studies suggested that parental education helps parents feel less anxious about injuring or causing harm to their children. Further, the association between the caregiver's age and HEPA was inconsistent [22, 34, 35, 39]. Only one study out of four indicated significant positive associations [34]. There was also strong evidence that marital status [22, 34, 35] and the household number of children were not related to HEPA among parents [22, 35, 37].

*Depression and anxiety.* Two quantitative studies showed significant negative associations between parental depression and anxiety, and HEPA [38, 42] and were complemented by one qualitative study [41]. These studies strongly evidenced that the severity of depression and anxiety symptoms were salient risks of non-HEPA. The only study where this association was not significant [34] is characterized by a weak methodological design (Table 1). This weakness can be traced to the sample predominated by fathers (51%) who are not usually involved in HEP [39] and a non-validated self-report measure of adherence [34].

*Stress and emotional burden/exhaustion.* Three articles highlighted that mothers' sense of insecurity and doubts about their ability to complete exercises and burnout level [34, 41], level of stress, and family problems [24] were detrimental to parents' HEPA.

*Positive cognitions*. Of 13 studies included in this review, five quantitative studies have investigated cognitive factors such as high self-efficacy, high knowledge and ability, and low perception of the barriers that enhance parents' HEPA [32, 35, 37, 41]. According to these studies, all these factors are positively correlated with HEPA.

*Social support*. Social support is the perception and experience that one is taken care of, that supportive resources are available from others, and, most commonly, that one is part of a social network that is supportive. Although social support may involve several types of support, it was only assessed through one item and identified as a potential factor enhancing the likelihood of HEPA in four quantitative studies evidenced [22, 35, 37, 41]. The same result was reported in one of the qualitative studies included in the review [43] which was carried out through individual, face-to-face, and semi-structured interviews.

*Family income and equipment*. There were mixed findings on the positive association between family income and adherence in five quantitative studies, with some studies indicating significant positive associations [24, 35, 40] and no significance [22, 39]. Also, the availability of the home equipment necessary for the exercises did not affect the level of parental HEPA [35, 37]. Despite few studies conducted, these results indicate that parents do not any special equipment to adhere to HEPA.

## Therapist guidance and advice

*Amount of exercise*. The number of exercises prescribed by health professionals was investigated by two studies: a quantitative study [24] and a qualitative study [44]. These studies highlighted that the more exercises prescribed, the less likely parents were to adhere to the recommendations. Healthcare professionals thus, need to take care of this while designing HEP for parents of C-CP.

*Exercises that improve outcomes*. There were consistent findings on the positive associations between parental perception of the benefits and efficacy of exercise and HEPA in five quantitative studies [24, 32, 35, 40, 41] and complemented by one qualitative study [44] which may support the generalisability of the findings. In this case, it is thought that parents are highly motivated and hopeful when their children's conditions improve thanks to exercises performed [40, 41].

*Exercise without adverse effects on children*. The perceived adverse effects or pain of the exercises on C-CP were found to be negatively related to HEPA in two quantitative studies [24, 42]. This finding was complemented by the qualitative study, which recognized pain as detrimental to HEPA [44]. Therefore, it can be concluded that the presence of pain or adverse effects during exercise plays a crucial role in executing and adhering to HEP.

*Easy and enjoyable exercises*. Three quantitative studies continually reported enjoyable and easy exercises [24, 41, 42] as potential factors of HEPA, demonstrating a largely positive relationship in multivariate analyses. This was also confirmed in one qualitative study [44]. As such, an effective home treatment program should contain easy, understandable, and entertaining exercises that are suited to daily activities.

*Physiotherapist's support*. The relationship between support from the physiotherapist and HEPA was investigated in seven quantitative studies [22, 32, 35, 37, 38, 41, 42] and a qualitative one [44]. There was mounting and consistent evidence that parents were more likely to adhere to home exercises when they were satisfied with the care from their therapist [35, 37, 38, 40] and informed about the progress and clarification of doubts [35, 37]. Further, parents who were motivated by the therapist advised to insert exercises into daily routines, assessed their skills, or asked about adherence at home by their therapists were more likely to comply with home exercises [22, 35, 37, 41, 42, 44]. Yet, some authors [45] have highlighted the risk that

this follow-up may lead to stress and feelings of guilt in parents. Similarly, Lillo-Navarro and his colleagues [37] suggested that healthcare providers must be aware of this circumstance to be perceived as a source of assistance rather than a judgemental bystander.

## Discussion

This systematic review uniquely focuses on paediatric home-based rehabilitation strategies, underscoring the indispensable role caregivers/parents play. Unlike previous reviews that concentrated on factors of adult patients' adherence to home-based [25] and clinical-based therapies [26], this review zeroes in on paediatric patients, particularly children with cerebral palsy (C-CP), and their caregivers. Prior studies have primarily evaluated adherence from the perspective of self-reliant adults, identifying factors such as low levels of physical activity, poor social support, low self-efficacy, a greater perceived number of barriers to exercise, helplessness, self-motivation, increased pain levels [26], depression, and anxiety during exercise as predictors of poor adherence [26]. However, the dynamics significantly shift when the focus turns to pediatric patients, where the caregiver's role becomes central to adherence.

To our knowledge, this is the first systematic review that objectively assesses the factors influencing adherence to home exercise programs among among caregivers, specifically mothers of C-CP. The evaluation of thirteen studies revealed a predominantly high quality of evidence. Our findings categorize the influencing factors HEPA into three main categories: child-related factors (such as gross motor function), parental-related factors (including education level, depression, and anxiety, stress, emotional exhaustion, high self-efficacy, high knowledge and ability, and low perception of the barriers, and social support), and therapist-related factors. Importantly, therapist-related factors extend beyond direct guidance and support to include the design and tailoring of exercise programs. This encompasses decisions regarding the exercise amount, type, and adjustments to avoid adverse effects, all of which are pivotal for enhancing adherence.

In the context of paediatric rehabilitation for C-CP, the caregiver's adherence to HEP is vital, given their significant role in executing and monitoring the prescribed exercises. The review highlights the importance of the physiotherapist's guidance and advice, not just in the traditional sense but as key in shaping the exercise program itself. Effective communication, education, and support from therapists significantly elevate a caregiver's capacity to adhere to HEP, particularly through enhanced understanding and skill acquisition [46]. Consistently, scholars revealed that when therapists focus on enhancing parents' knowledge and abilities by offering information [47], explaining the use of exercises [46], and using the child as an example, it improves adherence. It was also observed that informing parents about the benefits of exercise significantly boosts their chances of adhering to HEP, possibly due to a better understanding of the cause-effect relationship, which improves adherence rates, especially for activities that parents find more challenging to complete.

Moreover, our findings suggest a nuanced consideration for the types of exercises prescribed, as caregivers may exhibit varying levels of adherence depending on the exercise's perceived complexity and benefits. These findings align with previous studies indicating that most parents perceive flexibility exercises as unpleasant and complicated activities [48]. Therefore, when planning treatments with families, therapists should consider the impact of these types of exercises on adherence.

While the factor represented by gross motor function is a given condition that does not depend on subsequent interventions, the ability to perform HEP consistently can substantially improve patients' living conditions by recovering functions impaired by paralysis. With

paediatric patients, the performance of HEPA is strictly dependent on the caregiver's adherence to the physiotherapist's recommendations.

A noteworthy aspect of caregiver adherence emerges from the relationship and trust between the caregiver and the therapist. The caregiver's psychosocial well-being directly influences their capacity to engage with and adhere to HEP, underscoring the interdependence between the child's rehabilitation process and the caregiver's mental and emotional state. Recognizing this dynamic is essential for therapists in crafting interventions that are not only effective in achieving therapeutic goals but also in supporting the caregiver's well-being. Having a child with a disability places parents in the position of having to accept an 'unacceptable' situation. Parents are likely to be affected by their child's difficulties and suffering and are, therefore, more at risk of developing psychosocial problems that may, in turn, affect the well-being and rehabilitation process of children with cerebral palsy. Thus, the effectiveness of the physiotherapist's prescriptions depends on their ability to build a relationship of trust with the caregiver.

Our review attempts to bridge existing gaps in the literature by providing a comprehensive analysis of the factors affecting HEP adherence among caregivers of C-CP. It underscores the need to integrate "care "into the therapeutic equation, fostering a more holistic approach that considers the psychological, social, cultural of dimensions of rehabilitation [49]. This perspective challenges the conventional "cure versus care" dichotomy, advocating for a balanced focus that honours both the professional responsibility to cure and the family's role in providing care.

The deeper analysis into the discrepancies observed in the results section reveals that the diversity in study designs, population characteristics, and measurement tools used across the reviewed studies significantly influences these differences. Variability in methodologies for conceptualizing and measuring adherence—ranging from self-designed questionnaires in studies like Başaran et al., [34] and Rone-Adams et al., [24] to validated scales as used by Sel et al., [38]—affects the comparability of reported adherence rates. Furthermore, the demographic characteristics of the study populations, such as parents' gender, age, and cultural background, contribute to varying adherence levels. For instance, the study by Alwhaibi et al., [22] which involved a predominantly female sample from Saudi Arabia, contrasts with Gmmash et al., [32] where a significant number of participants were female from the USA. This variance highlights how cultural factors and parental roles within different societies may influence adherence levels, as suggested by the high compliance among Bedouin parents in Galil et al., [40] indicating potential cultural differences in perceptions of disability and therapy.

The age range and developmental stage of children varied significantly across studies, from early infancy to late childhood. Studies like Medina-Mirapeix et al., [35] focused on younger children (0.5–6 years), potentially influencing adherence differently compared to studies involving older children, due to factors such as parental motivation and the child's ability to participate actively in exercises. Notably, social support emerged as a critical factor in multiple studies, underscoring the importance of a supportive environment for enhancing adherence. The quality of the studies, varied, which could impact the reliability of the findings. Higher-quality studies like those rated with 9 stars on the quality measures provided more robust evidence compared to lower-rated studies. These elements collectively underscore the complex interplay between methodological approaches, population demographics, and cultural contexts in interpreting discrepancies among the studies reviewed, pointing to the necessity of a nuanced understanding of adherence within the specific context of home-based rehabilitation for children with disabilities.

## Study strengths and limitations

This review was rigorous and used a strong methodological approach to discover, appraise, and synthesize significant findings. The search parameters were intentionally broad to locate research examining a wide variety of factors that may impact HEPA among parents of children with cerebral palsy. It also used a robust quality criteria tool to assess the articles included in the synthesis.

Nevertheless, our findings should be interpreted considering the methodological limitations of included studies and the review itself. First, a small number of studies exploring several factors, studies' methodological constraints, changes in concept formulation and operationalization, and contradictions across findings have all made it difficult to draw firm conclusions regarding the predictive power of these factors. Second, the difficult measuring of home exercise adherence deserves special attention. As revealed in the previous reviews, the absence of verified and standardized methodologies is a significant methodological restriction in this field of research [25, 36], making it difficult to compare findings across studies.

Moreover, noting that all studies included in the syntheses applied the cross-sectional design, it's not possible to make conclusions about causal effects. Longitudinal studies are strongly recommended to draw definitive causal effects and confirm the presence or absence of associations. Lastly, only published full-text English language publications were considered, which may have created a publication bias given that unpublished studies may be more likely to have reported different findings. It might also be argued that the review did not consider differences in adherence metrics between studies sufficiently.

## Implications for research and practice

In terms of research, further studies are warranted to determine the nature of the relationships between HEPA and each of the five identified consistent predictors. It is crucial to understand whether certain groups of parents or specific circumstances amplify these predictors' effects, and to what extent other factors may mediate or moderate the relationship. Coupled with the expanding research on behaviour modification techniques [50], these findings may guide the development of highly effective strategies to boost HEPA. Identifying the most impactful adherence behaviour factors allows for a targeted approach in selecting of behaviour modification strategies for these interventions. Furthermore, despite acknowledging computer-based rehabilitation's increasing role in managing cerebral palsy and its influence on adherence, there is a noticeable gap concerning its psychosocial dimensions. This observation underscores the necessity for research focused on the psychosocial aspects of adhering to such rehabilitation modalities. Finally, future studies addressing the methodological limitations identified in this review, particularly the challenges in monitoring home-exercise adherence, is essential for enhancing the evidence base regarding HEPA predictors.

In terms of practice, the review's findings have implications for practitioners implementing HEP programs with parents of C-CP. Considering that stronger parental self-efficacy, high knowledge, and ability to perform the exercises, low perceived barriers, social support and therapist guidance, high gross motor function, and low psychological distress tend to predict higher HEP Adherence. Therefore, assessing these domains beforehand may uncover any risk factors for poor adherence. These findings give significant insights into how therapists and other stakeholders working with parents of C-CP may take a more active role in protecting against the alarming rate of HEP non-adherence by ensuring that appropriate assistance is in place or prioritizing currently accessible help. Priority areas should be reinforcing guidance of health professionals to parents, strong peer support networks, psychological support, and training of the parents to increase high self-efficacy, high knowledge, and ability to promote HEPA.

## Conclusion

This systematic review highlighted that the factors affecting HEPA among parents of C-CP are in three main categories such as child-related factors (gross motor function), parental-related factors (education level, depression, and anxiety, stress, emotional exhaustion, high self-efficacy, high knowledge and ability, and low perception of the barriers, and social support), and therapist's guidance and advice (amount of exercise prescribed, exercises that improve outcomes, exercise without adverse effects, and physiotherapist's support). Knowing and assessing these factors gives researchers and practitioners more options for improving adherence through intervention design and practices aimed at strengthening facilitators and reducing obstacles to adherence. The analysis also revealed that the determinants of HEPA are still not completely understood. The current review has recommended avenues for future research by highlighting inconsistent findings and methodological shortcomings.

## Supporting information

**S1 Checklist. PRISMA 2020 checklist.**
(DOCX)

## Acknowledgments

We would like to thank the Covidence team who provided free access to the Covidence, a screening and data extraction tool for conducting systematic reviews.

## Author Contributions

**Conceptualization:** Japhet Niyonsenga, Liliane Uwingeneye, Inès Musabyemariya, Jean Baptiste Sagahutu, Francesca Cavallini, Luca Caricati, Jean Mutabaruka, Stefan Jansen, Nadia Monacelli.

**Data curation:** Japhet Niyonsenga.

**Formal analysis:** Japhet Niyonsenga, Liliane Uwingeneye, Francesca Cavallini, Luca Caricati, Jean Mutabaruka, Stefan Jansen, Nadia Monacelli.

**Investigation:** Japhet Niyonsenga, Liliane Uwingeneye, Inès Musabyemariya, Jean Baptiste Sagahutu, Francesca Cavallini, Luca Caricati, Rutembesa Eugene, Jean Mutabaruka, Stefan Jansen, Nadia Monacelli.

**Methodology:** Japhet Niyonsenga, Inès Musabyemariya, Jean Baptiste Sagahutu, Francesca Cavallini, Luca Caricati, Rutembesa Eugene, Jean Mutabaruka, Stefan Jansen, Nadia Monacelli.

**Project administration:** Liliane Uwingeneye.

**Resources:** Luca Caricati.

**Software:** Jean Mutabaruka.

**Supervision:** Francesca Cavallini, Luca Caricati, Rutembesa Eugene, Jean Mutabaruka, Stefan Jansen, Nadia Monacelli.

**Validation:** Japhet Niyonsenga, Jean Mutabaruka.

**Writing – original draft:** Japhet Niyonsenga, Liliane Uwingeneye, Luca Caricati, Rutembesa Eugene.

**Writing – review & editing:** Japhet Niyonsenga, Liliane Uwingeneye, Inès Musabyemariya, Jean Baptiste Sagahutu, Francesca Cavallini, Luca Caricati, Rutembesa Eugene, Jean Mutabaruka, Stefan Jansen, Nadia Monacelli.

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
