## [Decision Letter · Decision Letter 0]

16 Feb 2024

PONE-D-23-41437The psychosocial determinants of adherence to home-based rehabilitation strategies in parents of children with cerebral palsy: A systematic reviewPLOS ONE

Dear Dr. Niyonsenga,

Thank you for submitting your manuscript to PLOS ONE. After careful consideration, we feel that it has merit but does not fully meet PLOS ONE’s publication criteria as it currently stands. Therefore, we invite you to submit a revised version of the manuscript that addresses the points raised during the review process.

**The paper reviews home-based rehabilitation strategies for parents of children with cerebral palsy, highlighting potential for improved adherence and program adaptation. However, limited research and methodological diversity constrain the study. Further detail on metrics and inclusion of graphical representations are recommended (see Reviewer#1 comments). While the narrative synthesis approach is suitable, addressing biases and study quality impact could enhance rigor. Acknowledged limitations include reliance on self-report measures and convenience sampling. The discussion should explore reasons for discrepancies among studies, including design and measurement variations (see Reviewer#2 comments).**

We look forward to receiving your revised manuscript.

Kind regards,

Jyotindra Narayan

Academic Editor

PLOS ONE

Journal Requirements:

3. In the online submission form, you indicated that this study included articles which are available via PubMed, Scopus, CINHAL, PsycINFO, and Embase. All information analysed in this study was collected in a dataset and this is available from the corresponding author on reasonable request.

Reviewers' comments:

Reviewer's Responses to Questions

**Comments to the Author**

1. Is the manuscript technically sound, and do the data support the conclusions?

Reviewer #1: Yes

Reviewer #2: Yes

2. Has the statistical analysis been performed appropriately and rigorously? 

Reviewer #1: N/A

Reviewer #2: Yes

3. Have the authors made all data underlying the findings in their manuscript fully available?

Reviewer #1: Yes

Reviewer #2: Yes

4. Is the manuscript presented in an intelligible fashion and written in standard English?

Reviewer #1: Yes

Reviewer #2: Yes

5. Review Comments to the Author

Reviewer #1: The paper presents a review of studies targeting home-based rehabilitation strategies in parents of children with cerebral palsy. The topic is interesting and might provide insights on how to increase adherence or adapt the program setting. However, the limited number of studies and the diversity of methodological approaches followed by each paper (which are also mentioned in the review) are limitations of the study. In addition, more details are required for some metrics and how they are measured, in particular, for metrics like self-efficacy or adherence. Please see my detailed comments below:

In the abstract, you define the term GMF (gross motor function), however, in the conclusion of the abstract, you use the term FGM. Do both terms describe gross motor function, or does the latter define something different?

In the introduction, the first sentence starts with the term CP; however, this abbreviation is introduced in the following sentence. Please address this issue.

In the introduction, you mention that the prevalence of CP in some countries is "1.8-2.3% per 1000 children," or in other words, 0.18‰-0.23‰. In the following sentence, you write that in Africa, the respective rate is "2-10 per 1000 live births." Do you mean 2-10% per 1000 children or 2‰-10‰?

C-CP is defined in the abstract. Please introduce this term in the main text as well.

Lines 90-91: Is this applicable only for families living in rural areas (e.g., involving parents in treatment decisions)?

Line 125: It is written that "The search terms were specially designed for the relevant database." Do you imply that the search terms were not the same for all the databases but differed across them? The terms mentioned in lines 126-131, are they connected with an AND or OR? Please provide a sample query you used for one database (e.g., PubMed). Did you also include terms starting with a prefix (e.g., adheren* which includes the terms adherence and adherent as keywords)?

Line 133: Please include a reference to the Covidence program.

Lines 135-138: Did you also include studies with computer-based rehabilitation for such children?

Line 143: Repetition related to Covidence.

Table and figures mentioned in the text (e.g., Table 1, line 156) are not available in the document or the PLOS ONE review system.

Lines 196-197: No study was conducted between 2005 and 2013?

Line 203: Please provide more details regarding the term adherence and how it is defined. Is it related to the "percent of suggested exercise done," or are exercise sessions done in a specific period of time? Does adherence change on a weekly basis, or is it defined as a single metric for the whole program duration? Does the duration of the program differ across the studies?

Lines 209-214: Given the issues related to the studies, how is it possible to have three studies with 9 stars (10 is the maximum grade)?

Line 208: The phrase "one was awarded 8 stars" is written twice.

Line 219: The sentence requires rephrasing.

In the section "Quality assessment of the included articles," more details related to the metrics or measures used in the studies (223-238) are needed.

Line 173: Please provide the number of the table.

Line 282: How was self-efficacy measured in those studies? The same for "low perception."

Line 260: The name of the section is "Caregiver-related factors," but parental-related factors are presented. Please explain.

In the results section, a description of the exercise sessions across the studies is needed to understand whether there are differences across the session settings. Thus, besides child/parental/therapist-related factors, program-related factors may exist too.

Reviewer #2: The manuscript presents a comprehensive review of factors influencing adherence to home exercise programs (HEP) for children with cerebral palsy, as perceived by their parents. The authors systematically searched databases up to March 2023 and included both quantitative and qualitative studies in their analysis. The findings are segmented into three main categories: child-related factors, caregiver-related factors, and therapist's guidance and advice. The review highlights the multifaceted nature of adherence to HEP, including the influence of the child's gross motor function, the caregiver's education level, depression, anxiety, self-efficacy, and the quality of therapist support.

Comments for Revision:

1. The manuscript could be strengthened by including graphical representations of the findings to better visualize the relationships between various factors and adherence. Including plots and diagrams can make the manuscript more impactful and is strongly recommended.

2. The narrative synthesis approach is well justified given the heterogeneity of the studies included. However, the authors might consider a more detailed explanation of how they addressed potential biases within the studies analyzed and the impact of study quality on the overall findings.

3. The paper acknowledges the limitations related to the quality of the included studies, such as the reliance on self-report measures and convenience sampling. It would be constructive to discuss how these limitations might affect the review's conclusions and what future research directions could mitigate these issues.

4. The results section highlights areas of agreement and disagreement among the studies reviewed. The discussion section could be strengthened by a deeper analysis of why these discrepancies might exist, considering factors such as study design, population differences, and measurement tools used.

5. Grammatical and typographical errors should be corrected to improve the readability of the manuscript.

6. PLOS authors have the option to publish the peer review history of their article (what does this mean?). If published, this will include your full peer review and any attached files.

Reviewer #1: No

Reviewer #2: **Yes: **SUBHASH PRATAP

---

## [Author Response · Author response to Decision Letter 0]

11 Apr 2024

We wish to express our sincere appreciation to the editor and reviewer for their insightful comments. We have done our best to address all the issues raised and we hope that the final version of this manuscript is much improved due to their comments. Attached is the rebuttal letter detailing the comments and their corresponding responses, and the changes in the manuscript.

---

## [Decision Letter · Decision Letter 1]

23 May 2024

PONE-D-23-41437R1The psychosocial determinants of adherence to home-based rehabilitation strategies in parents of children with cerebral palsy: A systematic reviewPLOS ONE

Dear Dr. Niyonsenga,

Thank you for submitting your manuscript to PLOS ONE. After careful consideration, we feel that it has merit but does not fully meet PLOS ONE’s publication criteria as it currently stands. Therefore, we invite you to submit a revised version of the manuscript that addresses the points raised during the review process.

The reviewers and editors have carefully examined the revsied manuscript and found them satisfactorily. There are a few minor corrections needed as mentioned by one of the reviewers. The authors are suggested to address them for better readability of the contributions.==============================

We look forward to receiving your revised manuscript.

Kind regards,

Jyotindra Narayan

Academic Editor

PLOS ONE

Journal Requirements:

Reviewers' comments:

Reviewer's Responses to Questions

**Comments to the Author**

1. If the authors have adequately addressed your comments raised in a previous round of review and you feel that this manuscript is now acceptable for publication, you may indicate that here to bypass the “Comments to the Author” section, enter your conflict of interest statement in the “Confidential to Editor” section, and submit your "Accept" recommendation.

Reviewer #2: All comments have been addressed

Reviewer #3: All comments have been addressed

2. Is the manuscript technically sound, and do the data support the conclusions?

Reviewer #2: Yes

Reviewer #3: Yes

3. Has the statistical analysis been performed appropriately and rigorously? 

Reviewer #2: Yes

Reviewer #3: Yes

4. Have the authors made all data underlying the findings in their manuscript fully available?

Reviewer #2: Yes

Reviewer #3: Yes

5. Is the manuscript presented in an intelligible fashion and written in standard English?

Reviewer #2: Yes

Reviewer #3: Yes

6. Review Comments to the Author

Reviewer #2: The author has revised the manuscript as per the reviewers recommendations. The manuscript can be accepted.

Reviewer #3: Thank you for submitting your work.

Here are my comments on the manuscript:

1) Introduction: Overall, please check citation formatting. Sometimes, there is space in between citation and text.

2) Material and methods: No comments.

3) Results: In table 1, there is no need to include title in first column, rather than that, I would suggest using the citation of particular paper. Also, there is too much text in table 1. Is it possible to divide table 1 into two tables or decrease the text in table 1.

4) Why are lines not numbered after line 251?

5) Title “Family Income and equipment” is not entirely in italics, is their reason for that.

6) Overall, please check your citations. In some of them, there is space and sometimes, there is not.

7) Please check for sentence errors in discussion. Line: “Therefore,, when planning treatments with families, therapists should consider their impact on adherence to these types of exercises”. Line: “It underscores the need to integrate “care “into the therapeutic equation, fostering a more holistic approach that considers the psychological, social, cultural of dimensions of rehabilitation”.

7. PLOS authors have the option to publish the peer review history of their article (what does this mean?). If published, this will include your full peer review and any attached files.

Reviewer #2: **Yes: **SUBHASH PRATAP

Reviewer #3: **Yes: **Alka Bishnoi

---

## [Author Response · Author response to Decision Letter 1]

24 May 2024

Thank you very much for the insightful comments raised from the first revision to this second revision. We have included in submission files the rebuttal letter composed of the comments and their corresponding responses.

---

## [Editor Report · Decision Letter 2]

30 May 2024

The psychosocial determinants of adherence to home-based rehabilitation strategies in parents of children with cerebral palsy: A systematic review

PONE-D-23-41437R2

Dear Dr. Niyonsenga,

We’re pleased to inform you that your manuscript has been judged scientifically suitable for publication and will be formally accepted for publication once it meets all outstanding technical requirements.

Kind regards,

Jyotindra Narayan

Academic Editor

PLOS ONE

Additional Editor Comments (optional):

The authors have addressed all the comments raised by the reviewers and editor. The manuscript is now being accepted. Congratulations to the authors for the quality work.
---

## [Editor Report · Acceptance letter]

3 Jun 2024

PONE-D-23-41437R2 

PLOS ONE

Dear Dr. Niyonsenga, 

I'm pleased to inform you that your manuscript has been deemed suitable for publication in PLOS ONE. Congratulations! Your manuscript is now being handed over to our production team.

Kind regards, 

on behalf of

Dr. Jyotindra Narayan 

Academic Editor

PLOS ONE